# Learn To Be Honest: Mitigate LLMs' Overconfidence for Improving Hallucination Detection with Self-Hesitation Activation

## Abstract

While Large Language Models (LLMs) have demonstrated strong performance across a wide range of natural language processing tasks, plausible but unfaithful content is still inevitably generated, which is known as factual hallucination. Previous methods, such as classifier training and uncertainty estimation, have been proposed for hallucination detection. However, it is widely found that LLMs express overconfidence by attempting to rationalize the incorrect outputs, leading to a misalignment between perceived uncertainty and knowledge boundary perception. It significantly undermines the effectiveness of existing hallucination detection methods. We study the correlation between hallucination and overconfidence, arguing that they are systematically inseparable in traditional training strategies due to overtraining. To address this, a series of analyses are conducted and a method called Self-Hesitation Activation Fine-Tuning (SHAFT) is proposed to align the uncertainty with the factual correctness, making LLMs "More Honest". Experiments demonstrate that our approach significantly mitigates the overconfidence of LLMs and decouples overconfidence with hallucination, making the nonfactual instances more distinguishable. Furthermore, evaluations across three benchmarks reveal that SHAFT greatly improves the performance of various hallucination detection methods before generation, consistently indicating its generalizability and computational efficiency.

## 1 Introduction

Although large language models (LLMs) have marked significant progress in various natural language processing tasks, the challenges of hallucination are still inevitable (Maynez et al., 2020b; Xu et al., 2024). LLMs occasionally generate nonsensical content that does not meet the expectations of the instructions. In real-world Question-Answering (QA) research, the generation of incorrect answers is approximately regarded as factual hallucination. Since hallucination leads the model to produce seemingly plausible but actually incorrect knowledge, it poses a great challenge for real-world applications, significantly bring discredit on the reliability of the model (Ji et al., 2023). Therefore, how to detect and mitigate hallucination, is necessary and has been widely studied (Zhang et al., 2023; Huang et al., 2025a).

To date, various mechanisms for hallucination detection have been proposed (Pan et al., 2025). Several ways for detection have been studied mainly. Some explicit approaches compare the generated output with the external knowledge sources, human evaluation, or a powerful verifier for fact verification (Yan et al., 2024; Manakul et al., 2023; Zhang et al., 2025b). But they are unstable and unreliable since the performances rely heavily on the correctness of the retrieved knowledge or the results of the verifier. In addition, some approaches generally fine-tune the LLMs to identify the potential hallucinations before generation (Kadavath et al., 2022; Kapoor et al., 2024; Burns et al., 2023), but a large scale of instances is necessary to be annotated for training, which is costly. Moreover, based on the close relationship between hallucinations and LLM uncertainty, an alternative method measures the implicit representation of output, such as the generation probability, hidden states, or distribution entropy (Ni et al., 2025; Azaria & Mitchell, 2023; Lee et al., 2025; Zhang et al., 2025a). Since such a series of approaches probes the internal of the LLMs, it is also called hallucination probing. Hallucination probing methods have achieved promising performance and been widely applied in recent years due to their effectiveness and low cost.

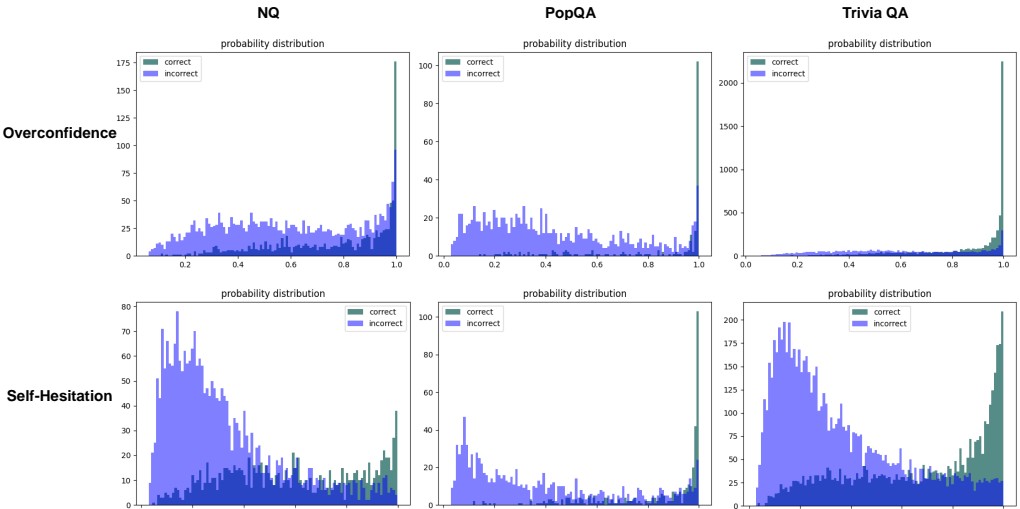

Figure 1: The figures demonstrate the distribution of the highest output probabilities among all the tokens in the vocabulary during generation, denoted as Maximum Token Probability (MTP) score in this paper. LLaMA3-8B-Instruct is utilized and three different datasets are evaluated, including Natural Questions (NQ) (Kwiatkowski et al., 2019), PopQA (Mallen et al., 2023), and Trivia QA (Joshi et al., 2017). The upper three figures showcase the distribution of the original model, exhibiting the overconfidence of the vanilla LLM. While the lower figures represent the model with self-hesitation activation, showing that LLM can honestly align the generated probability with correctness.

Despite the previous probing works about LLMs' uncertainty, we observe a key limitation that restricts the performance of the implicit hallucination detection methods: LLMs do not always exhibit uncertainty and low confidence for the erroneous outputs (Mielke et al., 2022; Fomicheva et al., 2020a; Xiong et al., 2024). In fact, a significant portion of generated misinformation is produced with high confidence, manifesting as overconfidence in the incorrect text. As shown in the first row of Figure 1, we discover that overconfidence widely appears in all benchmarks. The misalignment that widely exists between uncertainty and hallucination significantly violates the basic assumption of hallucination probing, where the factual hallucination is accompanied by uncertainty of the LLM generation. It will result in particular challenges for the existing hallucination probing methods.

However, in this paper, we find that the overconfidence that has been widely confirmed and discussed can actually be mitigated. Experiment results in the second row in Figure 1 demonstrate that LLMs potentially know the correctness of the generated content. We observe that a selective few-shot fine-tuning can easily make LLMs *more honest*. In this state, the overconfidence is greatly mitigated and the LLMs will honestly convey low probability when generating incorrect information, exhibiting uncertainty. We define this state that an LLM behaves more honestly as **self-hesitation** in this paper, where the LLMs actually know the correctness of their generation and learn to hesitate when generating uncertain information. However, systematic analysis has been adopted for further studies, revealing that self-hesitation is an intermediate state, which basically appears at the beginning of supervised fine-tuning and disappears as the fine-tuning continues. Several properties and causes of self-hesitation are concluded, making it more attributable.

Based on the discovery, we further leverage the analyses and propose **S**elf-**H**esitation **A**ctivation **F**ine-**T**uning (SHAFT), a novel training method to help hallucination detection, especially hallucination probing approaches. A selective and adaptive training is adopted for fine-tuning to stabilize the self-hesitation state, meanwhile enabling new knowledge learning. Subsequently, we leverage the properties of self-hesitation of LLMs for hallucination probing. It greatly mitigates overconfidence and promotes a closer correlation between uncertainty and hallucination, making the models more honest and the uncertainty more reliable. In this paper, six uncertainty estimation methods for hallucination probing, as well as two classifier-based probes, are introduced for analysis. Experiments conducted on two models and three benchmarks consistently demonstrate that almost all hallucination probing methods have improved significantly with self-hesitation. The considerable progress after SHAFT significantly demonstrates the effectiveness of our methods in hallucination detection and overconfidence mitigation.

Our main contributions are as follows:

- We systematically analyze the overconfidence of LLM that significantly disturbs hallucination detection and knowledge boundary perception, and first identify the state of self-hesitation as an intermediate state during supervised fine-tuning.

- A new training strategy called SHAFT is proposed to stabilize the self-hesitation. It effectively enables LLMs to learning new knowledge while maintaining self-hesitation.

- Results of eight different hallucination probing methods on two LLMs across three benchmarks consistently demonstrate the significance and effectiveness of our method.

## 2 RELATED WORK

**Hallucination Detection.** Since hallucinations are inevitable in LLM generation, previous works study how to detect hallucinations before output in case of generating incorrect or unrelated information (Pan et al., 2025). Some approaches verify the correctness and faithfulness of the text generated by LLM utilizing external knowledge sources, powerful verifiers, or human evaluation (Manakul et al., 2023; Yan et al., 2025; Zhang et al., 2025b). These methods require explicit information as the truths to verify the generated text, which is costly and unstable. Subsequent methods trained a classifier or fine-tuned LLMs to detect hallucinations while generating the responses (Maynez et al., 2020a; Ni et al., 2025; Kadavath et al., 2022). However, constructing the training datasets with extra label annotation and a large scale of binary discrimination training is required. Uncertainty estimation is also generally utilized for detection, such as direct prompting for black-box models and confidence probing for white-box models.

**Uncertainty Estimation.** Since the confidence of the LLM is considered closely related to the quality of the generated text, previous works exploit several uncertainty estimation approaches for downstream factuality tasks (Kotelevskii et al., 2025). According to the accessibility of LLMs, existing methods can be generally divided into two types: black-box methods and white-box methods (Vashurin et al., 2025). The white-box methods require access to the internal states of LLMs, while the black-box methods are restricted to the literal generated text only. Since open source models are utilized exclusively in this paper, we mainly study and discuss white-box methods.

**Hallucination Probing.** Previous studies find that LLMs mostly know what they know (Kadavath et al., 2022; Kapoor et al., 2024). When LLMs are trained and instructed to review the question, they can generally predict whether they can answer the question correctly. Subsequent works further develop the discovery by correlating hallucination with the uncertainty of LLMs (Fomicheva et al., 2020a). The correctness of the generated answer can be predicted by inspecting the implicit representations of LLMs. Given a white-box model, prior studies construct a lightweight probe to inspect the values of certain states and parameters in the models, and predict the uncertainty of LLMs to refer to the hallucinations (Ni et al., 2025; Zhang et al., 2025a). For instance, a series of work develops the potential relation between probability and uncertainty for hallucination probing (Lee et al., 2025; Burns et al., 2023). Hallucination is estimated with the probability of generated tokens, which is supposed to reflect the uncertainty of LLMs. However, it is significantly impacted by the overconfidence of LLMs, which is widely confirmed. Therefore, further studies tend to focus on the internal of the LLMs for a solution. Internal details, such as the logits and the output hidden states of certain layers, are generally selected for access and estimation (Vashurin et al., 2025; Fomicheva et al., 2020a; Darrin et al., 2023; Kuhn et al., 2023; Fomicheva et al., 2020b).

## 3 FINDINGS: SELF-HESITATION

Since the overconfidence of LLMs severely impacts the performance of hallucination detection, we discover a state of LLM defined as self-hesitation that can mitigate overconfidence. In this section, we specifically illustrate the discovery of self-hesitation as well as the analysis of its attributions and limitations as the preliminaries.

### 3.1 OVERCONFIDENCE

Based on previous works (Kadavath et al., 2022), we specifically inspected the uncertainty of LLMs when generating incorrect information. The Maximum Token Probability (MTP) score is mainly used for analysis due to its intuitive and convenient utilization. In this paper, we further explore the universality of overconfidence within current pre-trained LLMs. Experiments are conducted on

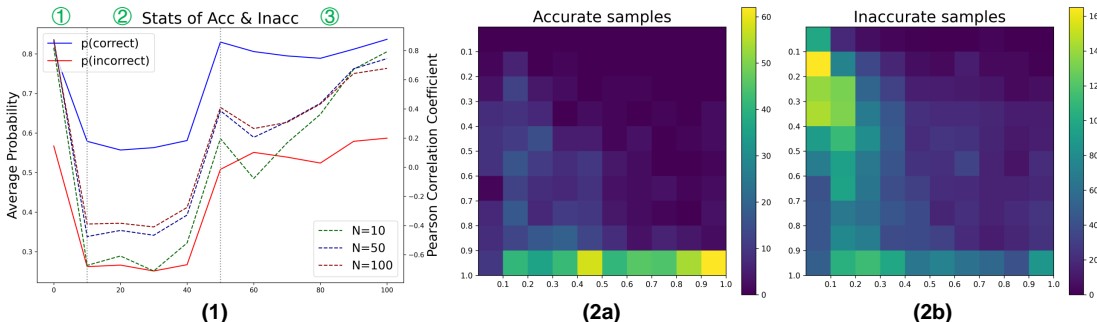

Figure 2: The figure demonstrates the change of the correct and incorrect samples as the SFT continues. Figure (1) showcases the relation between the probability distributions of the evaluated samples and the training steps. Specifically, the blue solid line means the average MTP score of correct samples, and the red one indicates the incorrect ones. The dashed lines demonstrate the Pearson Correlation Coefficient of MTP distributions between correct and incorrect samples. Figure (2a) and (2b) individually showcase the heatmaps of correct and incorrect samples about the change of MTP after self-hesitation. The demonstrated experiments are conducted on *LLaMA3-8B-Instruct*.

several LLMs, including LLaMA-3-8B-Instruct and Qwen2.5-7B-Instruct by directly providing the questions and ask for the straightforward answers. Results on various datasets, including PopQA, Natural Questions (NQ), and Trivia QA, consistently demonstrate that LLMs tend to output the key tokens with high probability regardless of their correctness. It leads to a great challenge for hallucination detection methods, which probe the uncertainty of LLMs during generation.

## 3.2 SELF-HESITATION ACTIVATION

Although supervised fine-tuning (SFT) can help mitigate hallucinations in LLMs, it often fails to address, and may even exacerbate the models' overconfidence. It makes the remaining hallucinations more difficult to detect. In this paper, we find that the issue can be immediately alleviated through a FEW examples through a simple SFT. A "Less is More" phenomenon (Zhou et al., 2023; Ye et al., 2025; Xiao et al., 2025) for uncertainty estimation is discovered, where only hundreds of selected examples can significantly mitigate the overconfidence and make a model "learn to be honest". The experiments in Figure 2 (1) present several metrics to quantify the overconfidence. It is obvious that during the fine-tuning, a sudden change is consistently observed at the first 10 steps. The confidence of almost all the instances, especially the incorrect ones, has significantly decreased. The drop in PCC scores further illustrates that most of the incorrect instances have much lower confidence in generation than the correct. Figure 2 (2a) and (2b) have demonstrated consistent results, indicating a significant distinction in behavior between both classes of samples. The confidence of an LLM is severely shaken after the SFT when a potential hallucination is generated, leading to hesitation in itself. Therefore, we define such a state, where LLMs honestly exhibit low confidence in incorrect generation and keep higher confidence in correct generation as **self-hesitation**. Based on the discovery and definition, we conduct a series of analyses and draw several key insights:

*First, the self-hesitation state is unstable during the natural SFT process.* In spite of the significant results of the experiments, we inspect that self-hesitation is solely the intermediate state during SFT, which is extremely unstable. As Figure 2 (1) shows, when the selective SFT continues, the LLMs will gradually exit the self-hesitation state and return to overconfidence once again. This finding leads to a conflict between knowledge learning and self-hesitation maintaining. If LLMs require to keep the state of self-hesitation, no more knowledge can be learned via SFT. Additionally, to further identify the particularity of the self-hesitation state, we compare the differences among the three states (original overconfidence, self-hesitation, and returned overconfidence in Figure 2 (1)) that occur during selective SFT, which is demonstrated in Appendix C.1.

*Second, hard samples are the main reason for activating the self-hesitation state rather than the easy ones.* We prompt the model to answer the questions in the training dataset first, separating the easy and hard samples according to the correctness of the results. Afterward, the model is trained with easy and hard samples individually in the same steps. Both models are evaluated, and the MTP

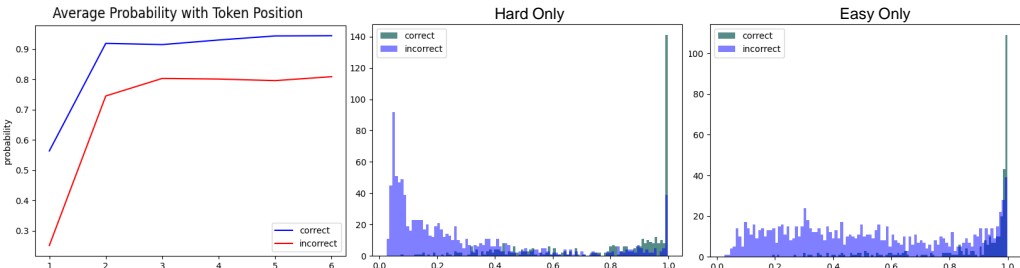

Figure 3: The figures demonstrate a further experimental analysis of the correlation between uncertainty and hallucination. The left figure shows the average probability of the tokens in different positions. The middle and right figures individually demonstrate the probability distribution of the evaluated instances that are trained with different samples. The model of the middle figure is only trained on hard samples that the model answers inaccurately originally, while the model of the right figure is the opposite. The demonstrated experiments are conducted on *LLaMA3-8B-Instruct*.

distributions are demonstrated in Figure 3. The self-hesitation is obviously observed in the hard training model, while the easy one still remains overconfident.

*Third, the uncertainty is mainly exhibited in the first tokens of the generated entities.* As demonstrated in Figure 3, the probabilities generally differ at the first tokens. As long as the first token is decided, the subsequent generation is determined with little hesitation, which meets the intuition that LLMs actually know what to say before generation. It also helps us to detect hallucination *before* generation in real-world applications, significantly preventing unfaithful generation and saving computational time. In addition, previous work (Ni et al., 2025) has found that a short-form response, which directly answers the question, would greatly outperform a chain-of-thought (COT) form of generation. Therefore, the instances are prompted with a short-form generation and the probability of the first token is sampled for uncertainty estimation in this paper.

In spite of the intuitive and remarkable performance of a selective SFT strategy for self-hesitation activation, where the predicted probability of LLMs can directly reflect potential hallucinations, existing shortcomings still can not be ignored. On the one hand, an inference on the training set first to filter out easy samples is required for a hard-only SFT, which is costly in data preparation. On the other hand, due to the instability of the self-hesitation, the number of training steps is uncontrollable, and the knowledge learning is challenging as well.

## 4 SHAFT: SELF-HESITATION ACTIVATION FINE-TUNING

In this paper, to leverage the discovery and avoid the limitations above, we develop a new training strategy named with **S**elf-**H**esitation **A**ctivation **F**ine-**T**uning (SHAFT). Inspired by the previous preliminaries and analysis, we aim to address the issues of model overconfidence and the instability inherent in self-hesitation states. We argue that conventional training strategies will gradually induce hallucination and overconfidence in models due to the generalization (Huang et al., 2025b). In this work, we focus primarily on supervised fine-tuning (SFT), a straightforward yet critical training approach, to investigate these phenomena. Specifically, the potential properties that can be blamed for hallucination are listed as follows:

First, over-training at the sample level is prevalent in previous strategies. Recent work (Kalai et al., 2025) has also suggested the potential causes of hallucination, where models might learn to favor confident guessing over acknowledging uncertainty. For a single sample during the standard training, models are typically optimized to maximize the logit scores of the annotated labels. However, it is sufficient for the target tokens to simply outperform other candidate tokens —excessively high confidence is unnecessary. Since the overall capabilities of a model are basically limited, redundant training on certain samples can lead to diminished performance in other domains or tasks. Therefore, we contend that ensuring the target tokens have higher probabilities than non-targets is adequate.

Second, the value of the training loss should account for the extent to which the target tokens are outperformed by the maximum-probability tokens. In conventional supervised fine-tuning, the loss depends only on the likelihood of the target token. If the target token's probability is close to the maximum, the corresponding loss should be substantially reduced. Therefore, we argue that the loss

should also reflect the divergence between the target token and the token with the current highest predicted probability. This adaptive strategy can effectively mitigate overtraining and substantially reduce the risk of the model returning to overconfident predictions.

To this end, we conclude the above issues and propose a selective and adaptive loss function to automatically select hard samples and modify the training loss. A filtering mechanism is introduced that exempts samples from further loss computation when the target tokens already achieve the highest probability among all candidates. Afterward, the loss function is weighted by the maximum token's probability. Finally, we propose the ultimate loss function of SHAFT as below:

$$\mathcal{L}_{SHAFT} = -\frac{\sum_1^N \mathbb{I}\left(P(\hat{y}_i \mid x_{<i}) \neq \max\{P(y_i \mid x_{<i})\}\right) \max\{P(y_i \mid x_{<i})\} \log P(\hat{y}_i \mid x_{<i})}{\sum_1^N \mathbb{I}\left(P(\hat{y}_i \mid x_{<i}) \neq \max\{P(y_i \mid x_{<i})\}\right)}, \quad (1)$$

where $N$ is the length of the generated sequence, $\mathbb{I}(\cdot)$ means that the value equals 1 only when the condition is satisfied; otherwise 0. Moreover, $\max\{P(y_i \mid x_{<i})\}$ indicates the maximum probability among all candidate tokens, and $P(\hat{y}_i \mid x_{<i})$ indicates the probability of the target token. Different from the direct hard fine-tuning, which is unstable during the entire training process, our SHAFT greatly stabilizes the status of self-hesitation, avoiding its premature disappearance.

## 5 EXPERIMENTS

We conducted experiments to extensively demonstrate the effectiveness of our hallucination probing methods. To illustrate the superiority and consistency, we compare our approach with other existing hallucination detection approaches. Then, we specifically analyze how self-hesitation improves the hallucination detection.

### 5.1 DATASETS, MODELS AND METRICS

**Datasets.** Three general and representative open-domain question-answering (QA) datasets are adopted in this paper, including Natural Questions (NQ) (Kwiatkowski et al., 2019), PopQA (Mallen et al., 2023), and TriviaQA (Joshi et al., 2017). NQ (Kwiatkowski et al., 2019) is a benchmark for QA research that contains real user questions issued to Google search, and answers found from Wikipedia by annotators. The open-domain subset NQ-open is utilized in this paper. PopQA (Mallen et al., 2023) is a short-form generation task where only one entity of factual knowledge is expected to be answered for each single question. In our experiments, we exactly followed the setting in the previous work, which evaluated methods on a long-tail subset consisting of 1,399 rare entity queries whose monthly Wikipedia page views are less than 100. TriviaQA (Joshi et al., 2017) is a reading comprehension dataset containing over 650K question-answer-evidence triples.

**Models.** Three white-box and open-source models are introduced for the experiments, including *LLaMA3-8B-Instruct* (Dubey et al., 2024) and *Qwen2.5-7B-Instruct* (Yang et al., 2024). The template of the input prompt for each model is demonstrated in Appendix B.1.

**Metrics.** Accuracy (Acc) is first utilized as an intuitive demonstration to measure the prediction performance. To avoid introducing hyperparameters in the experiments, the thresholds for discrimination are automatically set to the average score among all predictions. Additionally, following the previous studies of the binary classification tasks, Area Under Precision-Recall Curve (AUCPR) (Boyd et al., 2013) and Area Under Receiver Operating Characteristic (AUROC)(Bamber, 1975) are introduced to mitigate the impact of threshold selection and imbalanced datasets. specifically, AUCPR is an average of the precision weighted by the true positive rate of a given threshold $t$, which can be represented as:

$$AUCPR = \int_{-\infty}^{+\infty} P(h_{M,q} = 1 \mid \hat{h}_{M,q} > t) dP(\hat{h}_{M,q} \leq t \mid h_{M,q} = 1). \quad (2)$$

Meanwhile, AUROC is an average of the recall weighted by the false positive rate of a given threshold $t$, which can be calculated by:

$$AUROC = \int_{-\infty}^{+\infty} P(\hat{h}_{M,q} > t \mid h_{M,q} = 1) dP(\hat{h}_{M,q} \leq t \mid h_{M,q} = 0). \quad (3)$$

Table 1: Overall evaluation results on the test sets of three datasets. Results are separated based on the types of LLMs and are formalized as A/B, which individually represent the performance of the methods before/after SHAFT. **Bold** numbers indicate the performances that have improved with our SHAFT. Since too intensive **bold** numbers will not help emphasize the keynote, only the numbers in the Avg columns are **bold**.

| Methods | LLaMA3-8b-Instruct | | | | Qwen2.5-7b-Instruct | | | |
|---|---|---|---|---|---|---|---|---|
| | Acc | AUCPR | AUROC | Avg | Acc | AUCPR | AUROC | Avg |
| NQ | | | | | | | | |
| MTE | 58.8/64.8 | 87.1/90.3 | 74.7/80.9 | 73.5/**78.7** | 49.2/61.4 | 87.4/91.4 | 71.7/81.2 | 69.4/**78.0** |
| MTP | 61.8/72.7 | 85.3/88.7 | 70.6/78.8 | 72.6/**80.1** | 51.8/71.3 | 84.4/91.0 | 62.3/80.0 | 66.1/**80.8** |
| MSP | 49.2/64.9 | 81.2/91.1 | 69.4/82.0 | 66.6/**79.3** | 51.1/63.2 | 85.1/93.0 | 69.5/84.0 | 68.6/**80.1** |
| PPL | 58.9/65.3 | 87.3/90.4 | 73.9/81.1 | 73.4/**78.9** | 51.4/62.4 | 87.5/92.2 | 71.7/82.6 | 70.2/**79.1** |
| LS | 63.8/72.0 | 83.7/88.2 | 70.7/79.0 | 72.7/**79.7** | 58.6/73.6 | 85.4/90.8 | 66.8/80.5 | 70.3/**81.6** |
| FRD | 43.2/40.4 | 65.8/60.8 | 39.7/34.6 | 49.5/45.3 | 51.7/38.1 | 71.0/62.8 | 42.3/36.0 | 55.0/45.6 |
| PopQA | | | | | | | | |
| MTE | 55.7/56.8 | 95.3/84.3 | 83.3/75.6 | 78.1/72.2 | 44.1/65.7 | 94.3/94.5 | 72.3/88.2 | 70.2/**82.8** |
| MTP | 67.1/75.2 | 96.2/95.5 | 85.1/88.5 | 82.8/**86.4** | 58.6/76.6 | 97.3/96.5 | 86.2/91.3 | 80.7/**88.1** |
| MSP | 54.5/59.8 | 95.8/87.2 | 84.1/78.1 | 78.1/75.0 | 47.3/54.8 | 94.0/87.6 | 74.0/78.0 | 71.8/**73.5** |
| PPL | 55.6/59.2 | 95.1/86.4 | 81.5/77.1 | 77.4/74.2 | 46.3/67.1 | 95.7/95.2 | 78.2/88.8 | 73.4/**83.7** |
| LS | 66.9/73.5 | 95.7/93.0 | 84.0/83.8 | 82.2/**83.4** | 62.9/79.8 | 95.5/95.7 | 76.2/88.7 | 78.2/**88.1** |
| FRD | 40.6/49.1 | 74.8/75.8 | 25.5/51.2 | 47.0/**58.7** | 42.1/37.2 | 85.2/61.8 | 30.4/31.5 | 52.6/43.5 |
| TriviaQA | | | | | | | | |
| MTE | 75.6/65.2 | 77.6/86.5 | 81.5/79.2 | 78.2/77.0 | 63.2/63.3 | 81.1/86.7 | 77.4/76.6 | 73.9/**75.5** |
| MTP | 71.2/73.2 | 73.4/86.8 | 78.5/80.4 | 74.4/**80.1** | 64.6/74.4 | 78.9/89.2 | 73.5/80.1 | 72.3/**81.2** |
| MSP | 74.9/65.3 | 78.3/88.3 | 80.6/82.1 | 77.9/**78.6** | 63.2/63.9 | 80.2/88.8 | 78.1/79.9 | 73.8/**77.5** |
| PPL | 73.5/65.5 | 75.5/87.4 | 78.6/81.1 | 75.9/**78.0** | 64.2/64.3 | 81.7/88.9 | 77.9/79.9 | 74.6/**77.7** |
| LS | 75.7/74.6 | 78.4/85.6 | 82.0/79.8 | 78.7/**80.0** | 69.7/75.0 | 81.6/85.6 | 77.1/78.0 | 76.1/**79.5** |
| FRD | 32.5/36.3 | 31.6/55.1 | 27.4/35.3 | 30.5/**42.2** | 38.9/46.8 | 52.3/62.0 | 35.8/44.1 | 42.3/**51.0** |

## 5.2 Hallucination Probes

To demonstrate the effectiveness of self-hesitation activation intuitively, several easy but popular probes are introduced for hallucination detection. Following previous studies, we propose eight probes, which can be divided into two classes: calculator-based and classifier-based probes.

**Calculator Probes** detect hallucination by calculating the uncertainty of the model with the internal states. Several simple but typical uncertainty calculation methods are introduced in this paper, including (1) Mean Token Entropy (MTE) (Fomicheva et al., 2020a), which computes the average entropy of the tokens. (2) Maximum Token Probability (MTP), which collects the probability of the generated tokens. (3) Maximum Sequence Probability (MSP), one of the simplest methods that calculates the probability of the sequence of the generated tokens. (4) Perplexity (PPL, (Fomicheva et al., 2020a)), which computes the average negative log probability of generated tokens. (5) Lexical Similarity (LS, (Fomicheva et al., 2020a)), which calculates the average similarity scores among several responses. (6) Fisher-Rao Distance (FRD, (Darrin et al., 2023)), which computes the Fisher-Rao distance between the probability distribution and the uniform distribution. More details about the calculation of the probes can be found in Appendix B.2.

**Classifier Probes** build a lightweight multi-layer perception (MLP) network as the binary classifier for hallucination detection. The hidden states of the model are generally utilized as the input of the classifier (Ni et al., 2025; Zhang et al., 2025a). Previous studies (Ni et al., 2025; Azaria & Mitchell, 2023) have demonstrated the fact that the hidden states in the intermediate layers of models could involve more knowledge perception. Therefore, apart from the intuition that taking the last hidden states as the input, we also select the middle layer of the models for detection.

## 5.3 Results

Table 1 presents the results for all the models on three datasets. The performance of all the methods before and after self-hesitation activation can be directly compared through an intuitive observation. From these results, we can conclude the following findings:

Table 2: Accuracy results on the test sets of three datasets. Column Overall means the accuracy on the entire test set, while Column Balanced balances the proportion of positive and negative samples for the test. The accuracy results are formalized as A/B, which individually represent the performance of the methods before/after SHAFT. **Bold** numbers indicate the performances that have improved with our SHAFT.

| Methods | NQ | | PopQA | | TriviaQA | |
|---|---|---|---|---|---|---|
| | Overall | Balanced | Overall | Balanced | Overall | Balanced |
| LLaMA3-8b-Instruct | | | | | | |
| Calculator Voting | 64.4/**71.8** | 66.9/**74.1** | 64.4/**67.9** | 72.8/70.7 | 75.2/72.0 | 75.1/73.2 |
| Single Mid Layer | 69.8/**70.1** | 50.3/**51.2** | 69.7/**74.8** | 55.2/**58.9** | 65.7/60.7 | 61.1/**61.8** |
| Single Last Layer | 54.6/**68.6** | 57.2/53.2 | 60.2/**73.5** | 57.2/**59.4** | 52.8/**63.7** | 59.4/**62.5** |
| Qwen2.5-7b-Instruct | | | | | | |
| Calculator Voting | 59.3/**69.1** | 64.6/**72.7** | 60.0/**74.1** | 69.3/**79.7** | 68.3/**73.0** | 70.0/**72.9** |
| Single Mid Layer | 71.1/**72.8** | 55.1/51.7 | 73.6/**77.6** | 56.6/**64.2** | 48.9/**70.0** | 58.6/**67.2** |
| Single Last Layer | 64.9/**73.2** | 54.6/53.2 | 63.5/**78.1** | 57.1/**65.4** | 52.9/**72.8** | 60.6/**67.7** |

*First, in almost all proposed hallucination methods, the performances have significantly improved with our self-hesitation activation.* Specifically, as shown in Table 1, *LLaMA3-8b-Instruct* after SHAFT averagely outperformed the model before SHAFT by margins 5.2% on NQ with mean token entropy, 7.5% on NQ, 3.6% on PopQA and 5.7% on TriviaQA with maximum token probability, 12.7% on NQ and 0.7% on TriviaQA with maximum sequence probability, 5.5% on NQ and 2.1% on TriviaQA with perplexity, 7.0% on NQ, 1.2% on PopQA and 1.3% on TriviaQA with lexical similarity, and 11.7% on both PopQA as well as TriviaQA with Fisher-Rao distance. As for *Qwen2.5-7b-Instruct*, SHAFT averagely outperformed the original model by margins 4.5% on NQ, 12.6% on PopQA and 1.6% on TriviaQA with mean token entropy, 14.7% on NQ, 17.4% on PopQA and 8.9% on TriviaQA with maximum token probability, 11.5% on NQ, 1.7% on PopQA and 3.7% on TriviaQA with maximum sequence probability, 8.9% on NQ, 9.9% on PopQA and 3.1% on TriviaQA with perplexity, 11.3% on NQ, 9.9% on PopQA and 3.4% on TriviaQA with lexical similarity, and 8.7% on TriviaQA with Fisher-Rao distance. The advancements in our methods further emphasize the tremendous impact of overconfidence on hallucination probing and show the significance of mitigating the overconfidence with self-hesitation.

*Second, the classifier probing methods outperformed other baselines as well, revealing the consistent advancements of our SHAFT.* Since the classifier probes are trained solely for binary classification, no extra threshold is involved and it is not sensible to compare the AUROC and AUCPR, which are mainly introduced to eliminate the impact of the threshold. Therefore, we solely compare the accuracy of the hallucination detection. In addition, we combine the five calculator probes above to detect hallucination by majority voting. Over three among the approaches provide the positive results will be decided to be hallucination. As shown in Table 2, classifier-based probes with our SHAFT have also greatly outperformed the former ones, consistently illustrating the effectiveness of our methods.

### 5.4 Performance of Knowledge Learning and Hallucination Mitigation

Since supporting sufficient knowledge learning is one of the main contributions of our SHAFT compared with the direct hard-only fine-tuning, we analyze how the knowledge is learned and how the factual errors are mitigated. After over 1,000 steps of fine-tuning, our SHAFT still enables the models to be stabilized with self-hesitation, significantly outperforming the preliminary selective SFT, which loses the self-hesitation function with less than 50 steps. Meanwhile, our SHAFT can learn more new knowledge compared since far more instances are accessible during training. As

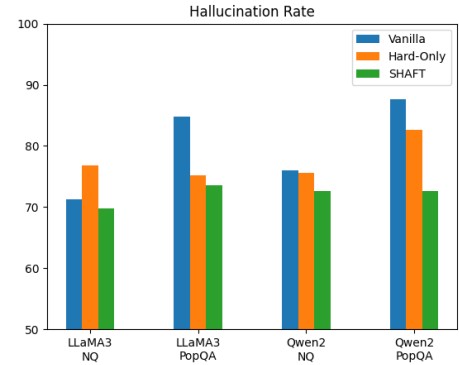

Figure 4: The overall hallucination rate (lower is better) on NQ and PopQA.

shown in Figure 4, the consistent improvements further indicate that SHAFT does not have much impact on knowledge learning, avoiding the trade-off between the self-hesitation function and new knowledge learning. Detailed results on TriviaQA are available in Appendix C.2 .

## 5.5 Curve of Receiver Operating Characteristic

In this section, we further study the performance of hallucination detection via the Receiver Operating Characteristic (ROC) curve. It assesses the diagnostic ability of a binary classifier by illustrating the true positive rate against the false positive rate. The performance is better when the curve is closer to the top-left corner. We adopt the ROC curve here for a more specific performance evaluation. Figure 5 demonstrates that uncertainty estimation methods have improved after SHAFT at any false positive rate. It indicates that no matter what the thresholds are set to, our SHAFT can consistently achieve a better performance. Since the limitation of the main text, more relevant results are available in Appendix C.3.

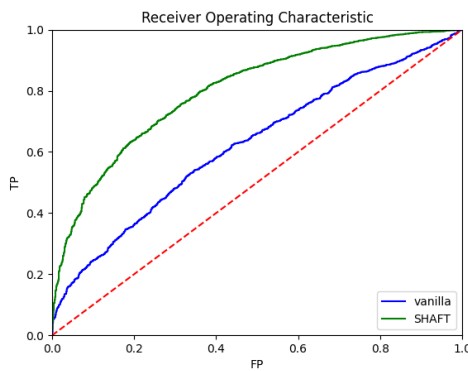

Figure 5: An example of the ROC curve of Maximum Token Probability on NQ with Qwen2.5.

## 5.6 Impact of Uncertainty with Long-Form Generation

As we illustrate above, previous studies have demonstrated that the performance of a COT-based long-form generation is greatly inferior to a short-form one (Ni et al., 2025). In this section, we present our experiment results to further support the proposition. As shown in Figure 6, The consistent superior performance of the short-form generation reveals its advancements in hallucination detection and knowledge perception. We suppose two reasons that mainly lead to these results. On the one hand, most of the generated tokens in the long-form responses are intermediate thinking content, lacking factual or key information. LLMs are generally supposed to be proficient in such form without much hesitation. Therefore, the methods that involve average uncertainty estimation are disturbed. On the other hand, a long-

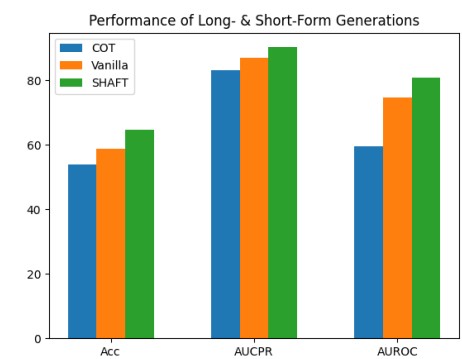

Figure 6: The performance of MTE with COT-form, vanilla short-form and SHAFT generation for LLaMA3-8b-Instruct on NQ.

form generation will enhance the confidence of the model to generate the ultimate answer, even if it is incorrect. It could intensify the overconfidence of the models and as a result, impair the effectiveness of hallucination detection. More comparison results are available in Appendix C.4.

## 6 Conclusion & Limitations

This paper discovers that the overconfidence of LLMs, which is generally inspected and assessed to impact the uncertainty of the models for hallucination detection, can be easily mitigated. Based on the analysis and experiments, the self-hesitation state of the LLMs is confirmed and a training strategy named Self-Hesitation Activation Fine-Tuning (SHAFT) is proposed. Training samples are selectively and adaptively fine-tuned based on their uncertainty and the self-hesitation state can be activated and stabilized during the knowledge learning. Subsequently, six calculator-based and two classifier-based probes are introduced for hallucination detection. Experiments on various models and benchmarks extensively demonstrate the consistency and effectiveness of our SHAFT method. Since SHAFT can make the LLMs more honest and build a strong correlation between uncertainty and hallucination, more applications besides hallucination detection can benefit as well, which have not been discussed in this paper and will be our future work.

## 7 ETHICS STATEMENT

We have checked and confirmed that the ICLR Code of Ethics is strictly followed. In addition, the LLMs and datasets are all open source and are utilized after confirming the licenses in this paper. Our proposed approach encourages the LLMs to be honest and avoid fabrication, aiming to improve the reliability of LLMs. No harmful induction exists in the experiments as well.

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

## A    THE USE OF LARGE LANGUAGE MODELS (LLMs)

We confirm that the Large Language Models are only used for grammar checking and text refining in context writing.

## B    EXPERIMENTS SETTING DETAILS

### B.1    PROMPT

> **Prompt for Short-Form Generation**
>
> Answer the question: QUESTION
> Answer as short as possible and generate the answer only.

> **Prompt for Long-Form Generation**
>
> Answer the question: QUESTION
> Think step by step and generate the answer at last.

### B.2    PROBES DETAILS

In this section, the detailed descriptions and calculations of the probes are listed below:

1. **Maximum Sequence Probability (MSP)** is one of the simplest methods, which computes the score:

$$U_{MSP} = 1 - \prod_{l=1}^{L} P(y_l \mid y_{<l}, x), \qquad (4)$$

   where $L$ is the length of the output tokens.

2. **Mean Token Entropy (MTE)** computes the entropy of softmax output distribution for all tokens and takes an average to obtain a sequence-level measure:

$$U_{MTE} = \frac{1}{L} \sum_{l=1}^{L} \mathcal{H}(y_l \mid y_{<l}, x) \qquad (5)$$

3. **Maximum Token Probability (MTP)** collects all the probability of the generated tokens. In the default setting, the selected tokens have the maximum probability in the entire vocabulary, which can potentially represent the uncertainty of the model. Following the previous experiments, the first token in the short-form generation is sampled for estimation.

4. **Perplexity (PPL)** method computes the average negative log probability of generated tokens. If the score is exponentiated, it corresponds to the perplexity:

$$U_{PPL} = \exp\left\{ -\frac{1}{L} \log P(y \mid x) \right\} \qquad (6)$$

5. **Lexical Similarity** calculates how similar two words or phrases are in terms of their meaning, which can be used for black-box models as well. It was originally proposed for machine translation. In this task, this measure iterates over all responses and calculates the average score with other answers.

6. **Fisher-Rao Distance (FRD)** computes the distance between the probability distribution for each token and the uniform distribution:

$$U_{FRD} = \frac{1}{L} \sum_{l=1}^{L} \frac{2}{\pi} \arccos \sum_{i=1}^{N} \sqrt{P(y_i \mid y_{<l}, x) \cdot \mathbf{q}_i}, \tag{7}$$

where N is the number of tokens in the vocabulary, and $\mathbf{q} = \left[\frac{1}{N}, ..., \frac{1}{N}\right]$ is a probability vector with a uniform distribution.

## B.3 DATASETS DETAILS

The detailed statistics of the datasets utilized in this paper are presented in Table 4.

Table 3: Statistics of the datasets.

| Dataset | Training Set | Test Set |
|---|---|---|
| NQ | 87,925 | 3,610 |
| PopQA | 12,749 | 1,399 |
| TriviaQA | 87,622 | 11,313 |

## B.4 TRAINING & INFERENCE DETAILS

We use the package *vllm* for inference, and the parameter settings are listed below:
```
temperature=0.0
top_p=1.0
max_tokens=100
skip_special_tokens=false
logprobs=5
```

The model was trained on 2*A800 in our experiment, and the SFT was implemented with the hyperparameter settings below:
```
n_step=1,000
batch_size=1
gradient_accumulation_steps=24
mixed_precision=bf16
max_seq_length=2048
warmup_ratio=0.02
learning_rate=2e-5
weight_decay=0.0,
```

## C SUPPLEMENTARY RESULTS

### C.1 CASE STUDY OF LAYER-LEVEL VARIATION BETWEEN OVERCONFIDENCE AND SELF-HESITATION MODEL

As Figure 2 shows, three main states are found. Therefore, we extract the checkpoints of the *LLaMA3-8B-Instruct* in all three states and further study the difference and observe how the hard-only SFT specifically works in the internal states at each layer. Three evaluation metrics are adopted for studying, including: (1) the Mean Squared Error (MSE) loss between the normalized hidden states of two models, (2) the KL divergence between the logits of two models, and (3) the predicted

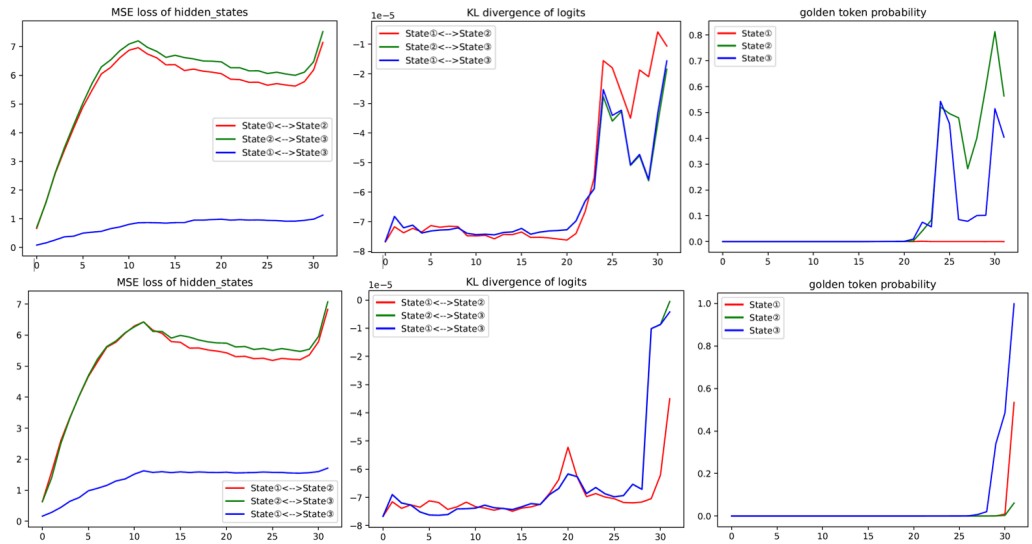

Figure 7: Case study of layer-level variation between overconfidence and self-hesitation model with *LLaMA3-8B-Insutrct*.

probability of the golden label. The logits and generating probabilities of the intermediate layers are calculated by extracting the hidden states for prediction. Case study is conducted in this section for detailed observation. As Figure 7 demonstrates that, the MSE of hidden states mainly arises in the middle and last few layers, which conform to the conclusion of previous studies (Ni et al., 2025; Azaria & Mitchell, 2023) as well. Specifically, the variation of hidden states mainly exists between State ① and State ②, ③, further revealing the particularity of the self-hesitation state.

## C.2 SUPPLEMENTARY EXPLANATION ABOUT TRAINING ON TRIVIAQA

Despite the improvements of both SFT and SHAFT on the overall accuracy across NQ and PopQA in Section 5.4, we find that any fine-tuning process in this paper can not make progress on TriviaQA. We suppose that the knowledge and domain may lack consistency, leading to the condition where more training is conducted, more hallucination is produced.

Table 4: Hallucination Rate performance on TriviaQA.

| TriviaQA | Vanilla | Hard-only SFT (30 steps) | Original SFT (1k steps) | SHAFT (1k steps) |
|---|---|---|---|---|
| LLaMA3-8B-Instruct | **42.7** | 77.0 | 56.1 | 64.7 |
| Qwen2.5-7B-Instruct | **56.6** | 78.8 | 59.6 | 67.8 |

## C.3 SUPPLEMENTARY ROC CURVES DEMONSTRATION

Considering the limitation of the main text, more supplementary ROC Curves are provided in this section along with Section 5.5.

### C.3.1 LLAMA3-8B-INSTRUCT

Figures 8, 9, 10, 11, and 12 demonstrate the results of the ROC curves on *LLaMA3-8B-Instruct*.

### C.3.2 QWEN2.5-7B-INSTRUCT

Figures 13, 14, 15, 16, and 17 demonstrate the results of the ROC curves on *Qwen2.5-7B-Instruct*.

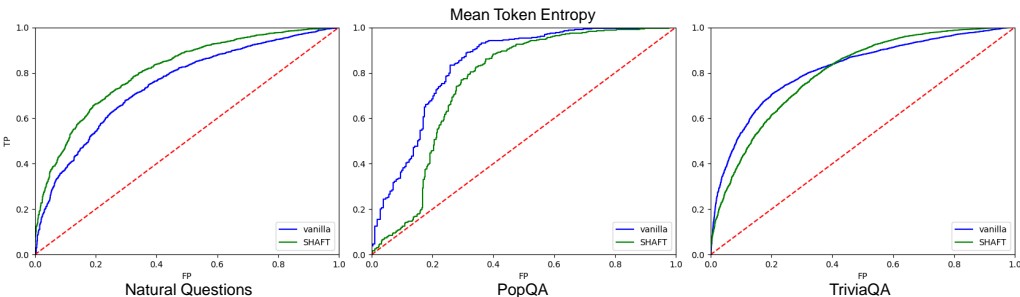

Figure 8: ROC curve of MTE uncertainty on benchmarks with *LLaMA3-8B-Insutrct*.

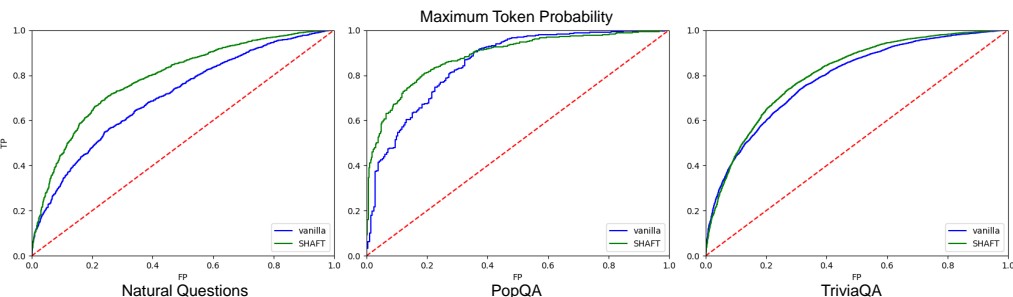

Figure 9: ROC curve of MTP uncertainty on benchmarks with *LLaMA3-8B-Insutrct*.

## C.4 SUPPLEMENTARY COMPARISON RESULTS OF LONG- & SHORT-FORM GENERATION

We provided more uncertainty estimation results of CoT-based generation in this Section along with Section 5.6.

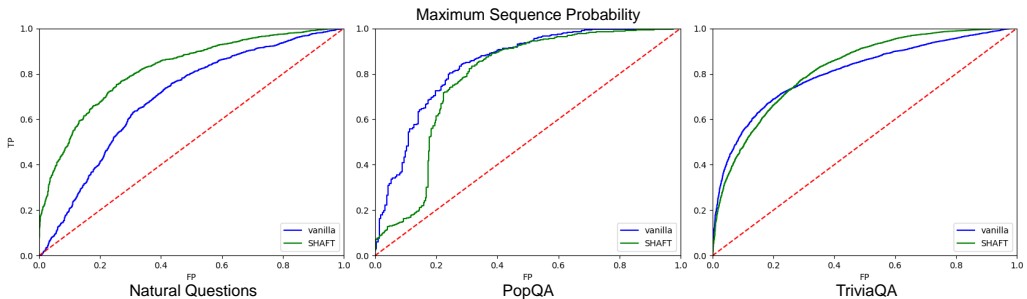

Figure 10: ROC curve of MSP uncertainty on benchmarks with *LLaMA3-8B-Insutrct*.

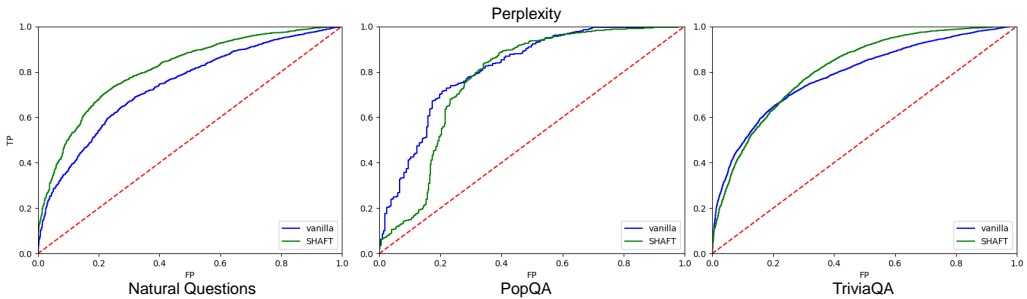

Figure 11: ROC curve of PPL uncertainty on benchmarks with *LLaMA3-8B-Insutrct*.

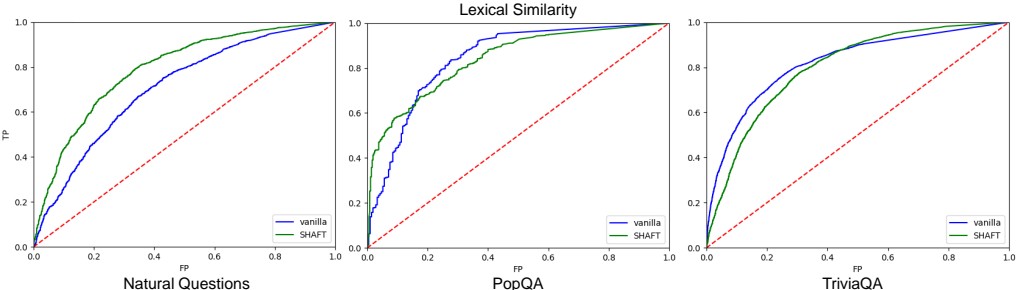

Figure 12: ROC curve of LS uncertainty on benchmarks with *LLaMA3-8B-Insutrct*.

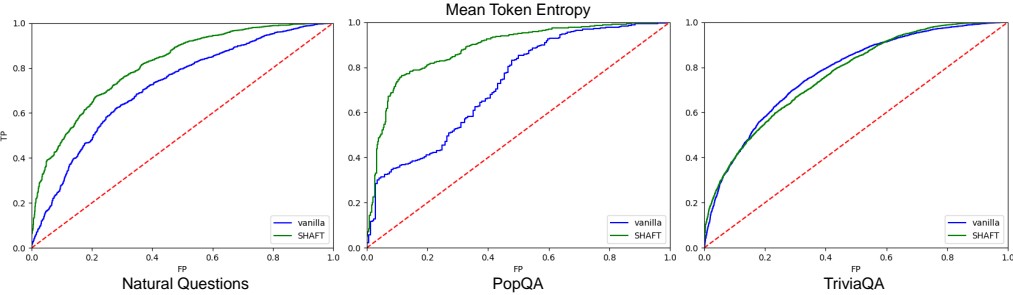

Figure 13: ROC curve of MTE uncertainty on benchmarks with *Qwen2.5-7B-Instruct*.

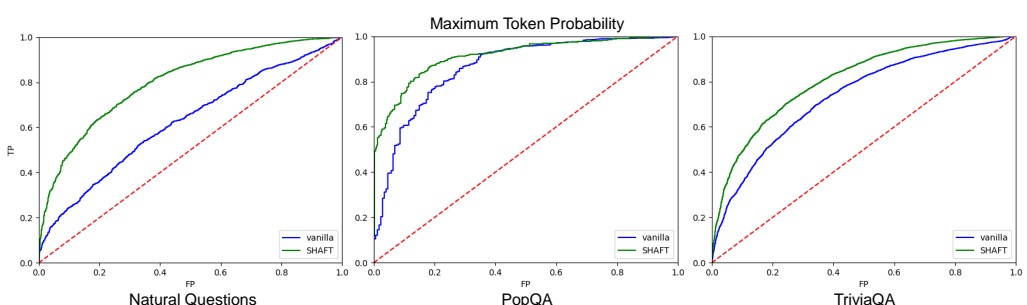

Figure 14: ROC curve of MTP uncertainty on benchmarks with *Qwen2.5-7B-Instruct*.

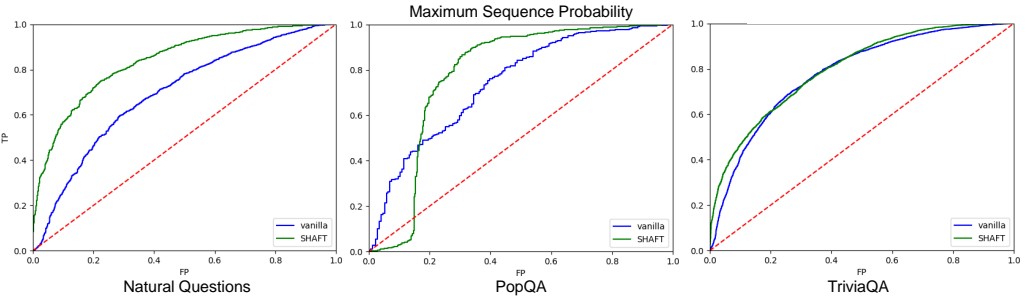

Figure 15: ROC curve of MSP uncertainty on benchmarks with *Qwen2.5-7B-Instruct*.

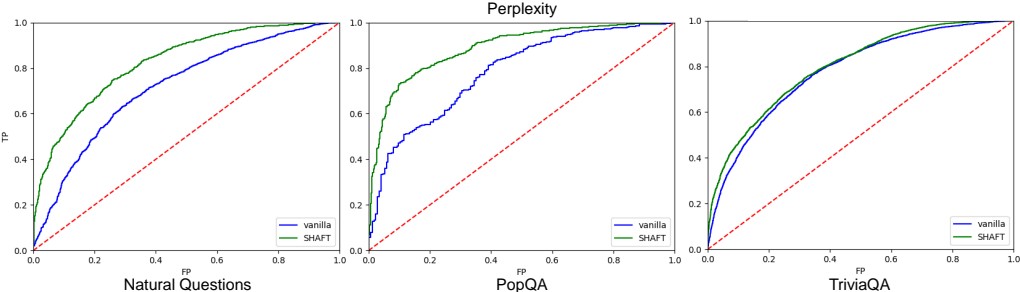

Figure 16: ROC curve of PPL uncertainty on benchmarks with *Qwen2.5-7B-Instruct*.

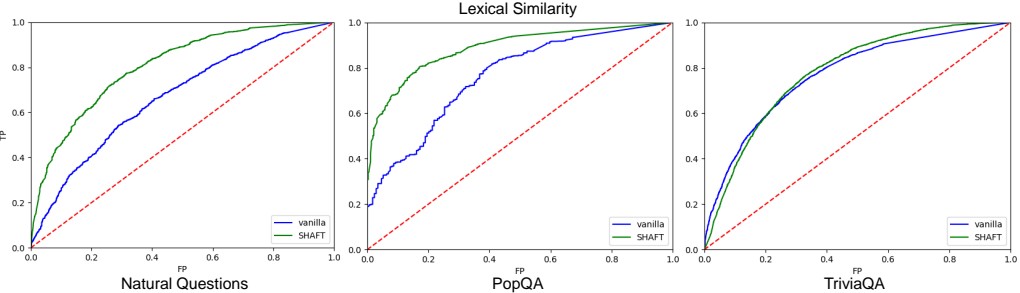

Figure 17: ROC curve of LS uncertainty on benchmarks with *Qwen2.5-7B-Instruct*.

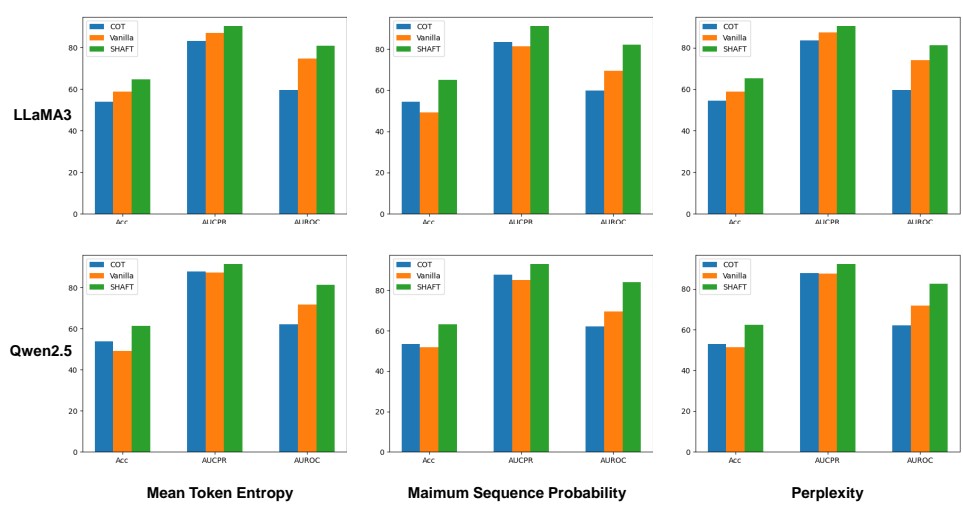

Figure 18: The performance of Uncertainty Estimation with COT-form, vanilla short-form and SHAFT generation for *LLaMA3-8b-Instruct* and *Qwen2.5-7B-Instruct*.

