# OpenReview forum: "Learn to be Honest: Mitigate LLMs' Overconfidence for Improving Hallucination Detection with Self-Hesitation Activation"
_ICLR.cc/2026/Conference — Submitted to ICLR 2026_

### Official Review · Reviewer_mehS · 2025-10-28

**Soundness:** 2
**Presentation:** 2
**Contribution:** 2
**Rating:** 2
**Confidence:** 4

**Summary:**

The paper addresses the problem that large language models (LLMs) often produce factually incorrect outputs while being overconfident, which undermines hallucination detection methods that rely on uncertainty signals. The authors identify a phenomenon they call self-hesitation, a transient state during fine-tuning where model confidence better aligns with correctness. They propose Self-Hesitation Activation Fine-Tuning (SHAFT), a fine-tuning strategy designed to stabilize this state. The method is evaluated across multiple datasets, two LLM families, and several baseline uncertainty estimation methods, showing improved alignment between confidence and correctness.

**Strengths:**

The paper presents an interesting observation on confidence propagation during supervised fine-tuning and proposes a novel method to mitigate model overconfidence. Experiments on multiple datasets demonstrate the effectiveness of the proposed approach.

**Weaknesses:**

(a) My major concern is the lack of comparison with standard SFT. While Table 1 compares the performance before and after the proposed fine-tuning method, the comparison is not fair because additional computation and extra data are utilized. To clearly demonstrate the benefit of the proposed approach, the baseline should include standard supervised fine-tuning (SFT) with cross-entropy loss.

(b) Averaging accuracy and AUROC in Table 1 is methodologically unsound — they measure fundamentally different aspects and are not directly comparable. This undermines the validity of some aggregated results and makes it unclear how much improvement truly comes from each dimension.

(c) The paper is generally understandable but would benefit from careful editing for grammar and phrasing. To name a couple:

Line 13: .. enables LLMs to learning new knowledge... --> to learn

Line  306:  Three white-box and open-source models are introduced for the experiments, including LLaMA3-8B-Instruct (Dubey et al., 2024) and Qwen2.5-7B-Instruct (Yang et al., 2024). --> the paper only presents results on two models.

**Questions:**

see weakness

---

> ### Author Response · Authors · 2025-11-21
>
> We thank the reviewer for the valuable suggestions!
> We will try our best to respond to the raised issues and will implement the suggested improvements in our revision.
>
> 1. Re: **Averaging the metrics is unsound**:
> > Thank you for your thoughtful review about the averaging scores. We agree that different metrics represent different aspects of a method's performance. However, on the one hand, as we illustrate in the caption of Table 1, too much score numbers, which related to different methods, different models and different metrics, can not help emphasize the keynote and make the table more clear and readable. We propose an average score just aiming to make the table more intuitive, which helps the readers directly view the variation of the overall performance. On the other hand, we just follow some previous researchers, who also selected the average of several metrics as an overall estimation. We think it may not be precise enough, but is valuable as a reference. Therefore, it is introduced in this paper, thank you for your advise, we eill consifer a better and balanced demonstration between the overall estimation and the individual metrics.
>
> 2. Re: **lack of comparison with standard SFT**:
> > Thank you for your feedback regarding the comparison with SFT as a baseline. Since our main purpose in this paper is to replace the standard SFT for a more honest learning paradigm, we admit that SFT is significant to be compared with.
> >
> > We consudcted a series of experiments on NQ around SFT and the following tables demonstrates the results:
> >
> > | LLAMA3-8B | MTE-Acc | MTE-AUCPR | MTE-AUROC | MTE-Avg |
> > | --- | --- | --- | --- | --- |
> > | Pre-trained | 58.8 | 87.1 | 74.7 | 73.5 |
> > | SFT         | 57.8 | 90.8 | 80.1 | 76.2 |
> > | SHAFT 　    | 64.8 | 90.3 | 80.9 | **78.7** |
> >
> > | LLAMA3-8B | MTP-Acc | MTP-AUCPR | MTP-AUROC | MTP-Avg |
> > | --- | --- | --- | --- | --- |
> > | Pre-trained | 61.8 | 85.3 | 70.6 | 72.6 |
> > | SFT         | 71.6 | 89.9 | 78.0 | 79.8 |
> > | SHAFT 　    | 72.7 | 88.7 | 78.8 | **80.1** |
> >
> > | LLAMA3-8B | MSP-Acc | MSP-AUCPR | MSP-AUROC | MSP-Avg |
> > | --- | --- | --- | --- | --- |
> > | Pre-trained | 49.2 | 81.2 | 69.4 | 66.6 |
> > | SFT         | 58.3 | 91.4 | 81.2 | 76.9 |
> > | SHAFT 　    | 64.9 | 91.1 | 82.0 | **79.3** |
> >
> > | LLAMA3-8B | PPL-Acc | PPL-AUCPR | PPL-AUROC | PPL-Avg |
> > | --- | --- | --- | --- | --- |
> > | Pre-trained | 58.9 | 87.3 | 73.9 | 73.4 |
> > | SFT         | 60.4 | 92.4 | 81.7 | 78.1 |
> > | SHAFT 　    | 65.3 | 90.4 | 81.1 | **78.9** |
> >
> > | LLAMA3-8B | LS-Acc | LS-AUCPR | LS-AUROC | LS-Avg |
> > | --- | --- | --- | --- | --- |
> > | Pre-trained | 63.8 | 83.7 | 70.7 | 72.7 |
> > | SFT         | 66.3 | 90.3 | 74.5 | 77.0 |
> > | SHAFT 　    | 72.0 | 88.2 | 79.0 | **79.7** |
> >
> > | Qwen2.5-7B | MTE-Acc | MTE-AUCPR | MTE-AUROC | MTE-Avg |
> > | --- | --- | --- | --- | --- |
> > | Pre-trained | 49.2 | 87.4 | 71.7 | 69.4 |
> > | SFT         | 58.0 | 90.8 | 76.5 | 75.1 |
> > | SHAFT 　    | 61.4 | 91.4 | 81.2 | **78.0** |
> >
> > | Qwen2.5-7B | MTP-Acc | MTP-AUCPR | MTP-AUROC | MTP-Avg |
> > | --- | --- | --- | --- | --- |
> > | Pre-trained | 51.8 | 84.4 | 62.3 | 66.1 |
> > | SFT         | 69.9 | 91.9 | 77.4 | 79.7 |
> > | SHAFT 　    | 71.3 | 91.0 | 80.0 | **80.8** |
> >
> > | Qwen2.5-7B | MSP-Acc | MSP-AUCPR | MSP-AUROC | MSP-Avg |
> > | --- | --- | --- | --- | --- |
> > | Pre-trained | 51.1 | 85.1 | 69.5 | 68.6 |
> > | SFT         | 59.4 | 93.4 | 81.7 | 78.1 |
> > | SHAFT 　    | 63.2 | 93.0 | 84.0 | **80.1** |
> >
> > | Qwen2.5-7B | PPL-Acc | PPL-AUCPR | PPL-AUROC | PPL-Avg |
> > | --- | --- | --- | --- | --- |
> > | Pre-trained | 51.4 | 87.5 | 71.7 | 70.2 |
> > | SFT         | 59.6 | 92.7 | 80.2 | 77.5 |
> > | SHAFT 　    | 62.4 | 92.2 | 82.6 | **79.1** |
> >
> > | Qwen2.5-7B | LS-Acc | LS-AUCPR | LS-AUROC | LS-Avg |
> > | --- | --- | --- | --- | --- |
> > | Pre-trained | 58.6 | 85.4 | 66.8 | 70.3 |
> > | SFT         | 71.9 | 90.7 | 76.4 | 79.6 |
> > | SHAFT 　    | 73.6 | 90.8 | 80.5 | **81.6** |
> >
> > The above results demonstrate that our SHAFT consistently outperform SFT, further illustrates the effectiveness of our methods. Since no any extra data annotation or preprocess is required compred with SFT, SHAFT can seamlessly replace the SFT in various applications.
>
> 3. Re: **editing for grammar and phrasing**:
> > Thanks for your reminding! We will carefully check the typos and errors once again and fix them in the revision.
>
> Please feel free to let us know if you have further questions. Thank you!

---

### Official Review · Reviewer_gqEm · 2025-10-30

**Soundness:** 3
**Presentation:** 3
**Contribution:** 3
**Rating:** 4
**Confidence:** 4

**Summary:**

This paper proposes an uncertainty-based training method called SHAFT. It applies adaptive weighting based on model confidence: for confident-but-incorrect predictions, it assigns high training weights; for uncertain predictions, it uses moderate weights; for correct predictions where the target token already has the highest probability, it skips training entirely. Experiments demonstrate strong performance improvements across multiple hallucination detection methods on three QA benchmarks.

**Strengths:**

1. SHAFT’s selective and adaptive training strategy makes sense. It adjusts the output distribution to better reflect the model’s actual knowledge boundaries rather than blindly reinforcing the pretrained model’s priors. It naturally aligns confidence with correctness. The method avoids overtraining on samples the model already knows while focusing on challenging cases.
2. The performance improvements are substantial and consistent, with gains of 5-17% across multiple hallucination detection methods, two different LLMs, and three benchmarks, demonstrating both effectiveness and generalizability.

**Weaknesses:**

The paper lacks baseline comparisons with similar SFT-based calibration methods: (1) Uncertainty-Aware Causal Language Modeling (UA-CLM) (Liu et al., 2024) which also modifies the loss function during fine-tuning to emphasize uncertainty for wrong predictions while optimizing for certainty on correct ones; (2) Unlikelihood Training (ULT) (Welleck et al., 2020) which adjusts training objectives by penalizing certain tokens. It would be better to have some comparison with previous methods.

**Questions:**

TriviaQA failure analysis: Why does SHAFT catastrophically fail on TriviaQA (hallucination rate increases from 42.7% to 64.7%, Table 4), while improving on NQ and PopQA? The explanation about "knowledge domain inconsistency" is vague. What specific characteristics of TriviaQA cause this? Can you provide case studies showing what goes wrong and whether this indicates fundamental limitations for certain question types?

---

> ### Author Response · Authors · 2025-11-21
>
> We thank the reviewer for the valuable and constructive suggestions and questions!
> We will try our best to respond to the raised issues and will implement the suggested improvements in our revision.
>
> 1. Re: **lacks baseline comparisons with similar SFT-based calibration methods like Uncertainty-Aware Causal Language Modeling (UA-CLM) and Unlikelihood Training (ULT)**
> > Thank you for your concerns and questions! We will discuss the two methods above individually. The paper "Neural Text Generation with Unlikelihood Training" designed an unlikelihood training (UL) to avoid "containing repeats and frequent words, unlike those from the human training distribution". The motivation and idea in this paper are totally different from our SHAFT, that aims to mitigate the overconfidence during the generation, which is not comparable.
> >
> > As for the paper "Enhancing Trust in Large Language Models with Uncertainty-Aware Fine-Tuning", who propose the UA-CLM to produce well-calibrated uncertainty estimates, design a loss function based on the token entropy. Compared with the paper, we have a more detailed experiments about the early exploration and discovery of vanilla SFT as a backup to stand for our theory. In fact, we have also studied the role of entropy, but decided not to involve entropy in the fine-tuning in the end. On the one hand, in our early study, we find that the token probability is a better signal for uncertainty detection. The distribution of the token probability (shown in Figure 1, Raw 2 Self-Hesitation) is much more distinctive and differentiated than the distribution of entropy. Although they have a correlation, a difference still exists. Since Figures are not allowed in this section, we will list them in the revision. Table 1 can also stand for our illustration, where MTP generally outperforms MTE in the same conditions. On the other hand, as the previous paper [1] reveals that, the tokens with high entropy generally denote the "fork" token but not key knowledge. Modifying the entropy in a large range may potentially affect the overall generalizability.
> >
> > Thanks for your suggestion again! Considering that UA-CLM is a valuable method for comparison, due to the limitation of the date and the lack of their released code, we will reproduce it and list the results later.
> >
> > [1] Beyond the 80/20 Rule: High-Entropy Minority Tokens Drive Effective Reinforcement Learning for LLM Reasoning, Arxiv.2506.01939
>
> 2. Re: **Question: analysis on TriviaQA**
> > Thank you for your question! First, we would clarify that since all TriviaQA, PopQA, and NQ construct questions from the open-domain corpus. The improvement on PopQA and NQ can reveal that the potential limitations for certain question types do not exist. Second, the consistent failure on SFT and hard-only SFT can prove that it does not come from our SHAFT as well. Advancements on PopQA and NQ stand for it as well. The only obvious difference between TriviaQA and the other two datasets is that TriviaQA is much bigger than PopQA and NQ. We only conducted 1,000 steps due to the limitations of the machines. Only 1,000 steps of fine-tuning may not completely make LLMs adapt to the TriviaQA benchmark (over 11k). Therefore, we think that is the main reason leading to the failure, which is the mentioned "knowledge domain inconsistency". Thanks for your question, we have clarified it and will add it in the revision and take more instances for fine-tuning if possible.
>
> Please feel free to let us know if you have further questions. Thank you!

---

> ### Author Response · Authors · 2025-12-01
>
> Re: **Reproduction and Comparison with Uncertainty-Aware Causal Language Modeling (UA-CLM)**
> > Since UA-CLM did not release the code in its work, we reproduced it and demonstrated the results on PopQA in the following table.
> >
> > | LLAMA3-8B | MTE-Acc | MTE-AUROC | MTP-Acc | MTP-AUROC | MSP-Acc | MSP-AUROC |
> > | --- | --- | --- | --- | --- | --- | --- |
> > | Pre-trained | 58.8 | 74.7 | 61.8 | 70.6 | 49.2 | 69.4 |
> > | SFT         | 57.8 | 80.1 | 71.6 | 78.0 | 58.3 | 81.2 |
> > | UA-CLM      | 53.3 | 67.2 | 65.8 | 25.2 | 35.9 | 26.4 |
> > | SHAFT       | **64.8** | **80.9** | **72.7** | **78.8** | **64.9** | **82.0** |
> >
> > It is obvious that our SHAFT significantly outperforms UA-CLM in all benchmarks, especially in Maximum Token Probability and Maximum Sequence Probability.
> >
> > Furthermore, following the loss function of UA-CLM:
> >
> > $L_{\text{UA-CLM}} = {-\frac{1}{|\widetilde{C}|}\sum_{i \in \widetilde{C}} P(w_i|w_{0:i-1}) \log (\tanh(H_i))} - {\frac{1}{|{C}|}\sum_{i \in {C}} (1-P(w_i|w_{0:i-1})) \log (1-\tanh(H_i))}$,
> >
> > where $H_i$ denotes the entropy of the i-th token probability distribution $\widetilde{C}$ and $C$ denotes the set of incorrect and correct tokens, it can be seen that the method does not take the ground-truth token fitting as the training target, which is totally different from SFT. UA-CLM only attempts to raise the uncertainty of the incorrect tokens and lower it in the correct ones, but ignores that the overall performance of the LLMs will be significantly affected through the training. Our reproduction also demonstrates the collapse in performance, revealing its potential challenges in real-world applications compared with our SHAFT.

---

### Official Review · Reviewer_Qzdg · 2025-11-01

**Soundness:** 3
**Presentation:** 3
**Contribution:** 2
**Rating:** 4
**Confidence:** 3

**Summary:**

This paper proposes Self-Hesitation Activation Fine-Tuning (SHAFT), a training method that mitigates LLM overconfidence by stabilizing an intermediate "self-hesitation" state where models honestly express low confidence when generating incorrect outputs, thereby improving hallucination detection performance.

**Strengths:**

1. Effective calibration approach: This work calibrates LLM outputs by aligning confidence with correctness, thereby enhancing the reliability of model predictions.
2. Simple and accessible methodology: The proposed method is straightforward to understand and implement, requiring only minimal modifications to standard training procedures.

**Weaknesses:**

1. Similarity to prior calibration methods: The core idea overlaps significantly with previous works "Alignment for Honesty" (training models to honestly express uncertainty) and "R-tuning" (selective training on hard samples for calibration). The paper does not clarify how SHAFT differs or what specific advantages its loss formulation provides over these existing approaches.
2. Missing comparisons: No empirical comparisons or ablations against these related methods are provided, making it unclear whether improvements stem from SHAFT's novel aspects or simply from applying known principles.

**Questions:**

1. What specifically differentiates SHAFT from "Alignment for Honesty" and "R-tuning"? Please clarify the conceptual and methodological distinctions beyond the specific loss formulation.
2. Can you provide empirical comparisons? If the core ideas are similar, direct performance comparisons with these baseline methods would help establish SHAFT's advantages.


[1]. Yang Y, Chern E, Qiu X, et al. Alignment for honesty[J]. Advances in Neural Information Processing Systems, 2024, 37: 63565-63598.
[2]. Zhang H, Diao S, Lin Y, et al. R-tuning: Instructing large language models to say ‘i don’t know’[C]//Proceedings of the 2024 Conference of the North American Chapter of the Association for Computational Linguistics: Human Language Technologies (Volume 1: Long Papers). 2024: 7106-7132.

---

> ### Author Response · Authors · 2025-11-21
> **Response 1**
>
> We thank the reviewer for the valuable suggestions and questions!
> It is exciting to receive many constructive concerns and suggestions.
> We will try our best to respond to the raised issues and will implement the suggested improvements in our revision.
>
> 1. Re: **Similarity to previous works "Alignment for Honesty" and "R-tuning"** & **Question 1: What specifically differentiates SHAFT from "Alignment for Honesty" and "R-tuning"**
> > Thanks for your valuable question! We would clarify that our method is fundamentally different from these methods according to the target of the training, which is not emphasized in the paper.
> >
> > We did notice that many previous works focus on how to train a model to specifically express refusal or IDK (I don't know). As we illustrate in the second paragraph in the Introduction, some approaches fine-tuned the LLMs to identify the hallucination or unknown knowledge during generation, which fits the situations of the two approaches above. They are great studies and we will add them to our reference. However, first, they generally require massive data preprocessing or annotations, which is costly for various applications and affects their efficiency. Second, some methods introduce some special prompt templates or special tokens to guide the LLMs to "speak" uncertainty (such as p(True), etc). It means that the methods are designed to dedicate fine-tuned models to hallucination detection tasks, potentially affecting their ability and generalizability on other tasks. Third, the purpose of these methods, like  "Alignment for Honesty" and "R-tuning", is to directly show LLMs what is correct and what is incorrect. During the processing, LLMs learn to "say" uncertainty but not "feel" uncertain.
> >
> > As a comparison, SHAFT **does not require more extra annotations or preprocessing than SFT**. It indicates that as long as the training data can be used for SFT, it can be seamlessly and directly utilized for SHAFT. Second, we would emphasize that our method does not teach LLMs to "speak" "I don't know", but inspire/activate LLMs to **"feel" uncertain** during answering through a knowledge-leanring fine-tuning like SFT. From this point, we suppose that our SHAFT is more similar to SFT than the methods above.
> >
> > As a conclusion, in fact, we just want to design a new SFT strategy to avoid overconfidence, making models more honest through SFT. After SHAFT, the LLMs can also be utilized to the methods you mentioned for further enhancement, which is not conflicting. Thank you again for your question, and we will add it to our revision!

---

> ### Author Response · Authors · 2025-11-21
> **Response 2**
>
> 2. Re: **Lack of empirical comparisons or ablation studies**
> > Thanks for your suggestions! Since our main contribution is finding a key nature of LLMs about uncertainty expression during a selective SFT, as well as designing a new SFT strategy based on the discovery, the most similar and comparable baseline is SFT. We conducted a series of experiments on NQ around SFT, and the following tables demonstrate the results. We will complete the experiments and add them to the revision. Also, thank you for your suggestion on the two papers, which will be cited in this paper as well. Limited by the date, we will consider reproducing them and add them to the analysis section later.
> >
> > | LLAMA3-8B | MTE-Acc | MTE-AUCPR | MTE-AUROC | MTE-Avg |
> > | --- | --- | --- | --- | --- |
> > | Pre-trained | 58.8 | 87.1 | 74.7 | 73.5 |
> > | SFT         | 57.8 | 90.8 | 80.1 | 76.2 |
> > | SHAFT 　    | 64.8 | 90.3 | 80.9 | **78.7** |
> >
> > | LLAMA3-8B | MTP-Acc | MTP-AUCPR | MTP-AUROC | MTP-Avg |
> > | --- | --- | --- | --- | --- |
> > | Pre-trained | 61.8 | 85.3 | 70.6 | 72.6 |
> > | SFT         | 71.6 | 89.9 | 78.0 | 79.8 |
> > | SHAFT 　    | 72.7 | 88.7 | 78.8 | **80.1** |
> >
> > | LLAMA3-8B | MSP-Acc | MSP-AUCPR | MSP-AUROC | MSP-Avg |
> > | --- | --- | --- | --- | --- |
> > | Pre-trained | 49.2 | 81.2 | 69.4 | 66.6 |
> > | SFT         | 58.3 | 91.4 | 81.2 | 76.9 |
> > | SHAFT 　    | 64.9 | 91.1 | 82.0 | **79.3** |
> >
> > | LLAMA3-8B | PPL-Acc | PPL-AUCPR | PPL-AUROC | PPL-Avg |
> > | --- | --- | --- | --- | --- |
> > | Pre-trained | 58.9 | 87.3 | 73.9 | 73.4 |
> > | SFT         | 60.4 | 92.4 | 81.7 | 78.1 |
> > | SHAFT 　    | 65.3 | 90.4 | 81.1 | **78.9** |
> >
> > | LLAMA3-8B | LS-Acc | LS-AUCPR | LS-AUROC | LS-Avg |
> > | --- | --- | --- | --- | --- |
> > | Pre-trained | 63.8 | 83.7 | 70.7 | 72.7 |
> > | SFT         | 66.3 | 90.3 | 74.5 | 77.0 |
> > | SHAFT 　    | 72.0 | 88.2 | 79.0 | **79.7** |
> >
> > | Qwen2.5-7B | MTE-Acc | MTE-AUCPR | MTE-AUROC | MTE-Avg |
> > | --- | --- | --- | --- | --- |
> > | Pre-trained | 49.2 | 87.4 | 71.7 | 69.4 |
> > | SFT         | 58.0 | 90.8 | 76.5 | 75.1 |
> > | SHAFT 　    | 61.4 | 91.4 | 81.2 | **78.0** |
> >
> > | Qwen2.5-7B | MTP-Acc | MTP-AUCPR | MTP-AUROC | MTP-Avg |
> > | --- | --- | --- | --- | --- |
> > | Pre-trained | 51.8 | 84.4 | 62.3 | 66.1 |
> > | SFT         | 69.9 | 91.9 | 77.4 | 79.7 |
> > | SHAFT 　    | 71.3 | 91.0 | 80.0 | **80.8** |
> >
> > | Qwen2.5-7B | MSP-Acc | MSP-AUCPR | MSP-AUROC | MSP-Avg |
> > | --- | --- | --- | --- | --- |
> > | Pre-trained | 51.1 | 85.1 | 69.5 | 68.6 |
> > | SFT         | 59.4 | 93.4 | 81.7 | 78.1 |
> > | SHAFT 　    | 63.2 | 93.0 | 84.0 | **80.1** |
> >
> > | Qwen2.5-7B | PPL-Acc | PPL-AUCPR | PPL-AUROC | PPL-Avg |
> > | --- | --- | --- | --- | --- |
> > | Pre-trained | 51.4 | 87.5 | 71.7 | 70.2 |
> > | SFT         | 59.6 | 92.7 | 80.2 | 77.5 |
> > | SHAFT 　    | 62.4 | 92.2 | 82.6 | **79.1** |
> >
> > | Qwen2.5-7B | LS-Acc | LS-AUCPR | LS-AUROC | LS-Avg |
> > | --- | --- | --- | --- | --- |
> > | Pre-trained | 58.6 | 85.4 | 66.8 | 70.3 |
> > | SFT         | 71.9 | 90.7 | 76.4 | 79.6 |
> > | SHAFT 　    | 73.6 | 90.8 | 80.5 | **81.6** |
> >
> > The above results demonstrate that our SHAFT consistently outperforms SFT, further illustrating the effectiveness of our methods. We will add it to our revision.
>
> We split the comment into two parts due to the length limit. Please feel free to let us know if you have further questions. Thank you!

---

### Meta-Review · Area_Chair_QHjM · 2026-01-07

**Summary:**

This paper proposes Self-Hesitation Activation Fine-Tuning (SHAFT)  that is kinda like a soft version of hard fine-tuning losses. This fine-tuning method can directly replace traditional SFT, and intuitively, it only fine-tunes tokens which are not too confident and weight the per-token loss by the uncertainty -- uncertain target tokens will get larger weights and the authors expect this approach to mitigate overconfidence and reduce hallucination. The reviewers are concerned about the novelty and missing baselines, and particularly, the authors didn't compare with a vanilla SFT baseline in the original submission.

During the rebuttal, the authors have added the SFT results that should address the concerns on it. However, overall, I think the proposed approach is interesting but the experiments evidence is not very convincing for me. As a potentially neat and effect method to directly replace SFT to reduce hallucination, I am not very convinced by just NQ, PopQA, and trivia QA results at this timing. I suggest at least the authors should demonstrate in general alignment like in chatting scenarios, to show whether hallucination can be really mitigated.

**Reviewer Concerns:**

Concerns addressed by rebuttal:

1. Missing standard SFT baseline: The authors added direct comparisons against standard SFT in the rebuttal, and the added results consistently show SHAFT outperforming SFT for both LLaMA3-8B and Qwen2.5-7B. This concern is largely resolved I think.
2. Comparison to UA-CLM: In response to the request for related baselines, the authors report having reproduced UA-CLM and claim it performs substantially worse (and can even collapse) compared to SHAFT. This partially addresses “compare to similar loss-modification fine-tuning” requests.
3. Clarification of relationship to “honesty / IDK” training: I don't think this is an issue as the authors explained.

Concerns that are still outstanding:

1. Breadth and quality of baseline coverage: UA-CLM is addressed, but other requested comparisons (e.g., to “honesty alignment” / “R-tuning”-style approaches, or other SFT-based calibration strategies) are not yet comprehensively benchmarked. While I agree the authors that these methods are not directly comparable since they typically require additional annotations, one could simply use confidence as the authors have used to determine uncertainty and train the models to predict "IDK" rather than modifying per-token losses. This could be a baseline of R-tuning.
2. There are some presentation issues complained, e.g., I agree with the reviewer that different metrics are not sound to be directly averaged.

**Reviewer Scores:**

1. Reviewer Qzdg (score 4->4): I think the authors did a good job clarifying the difference with R-tuning, but the authors didn't do further benchmarking with methods like that.

2. Reviewer gqEm (score 4 → likely 6): They requested comparisons to UA-CLM and (optionally) ULT, plus an explanation for the TriviaQA failure. The authors addressed UA-CLM with a reproduction and gave a concrete (though limited) explanation for TriviaQA.

3. Reviewer mehS (score 2 → likely 2 or 4)
This reviewer’s strongest issue was the lack of a fair SFT baseline; the rebuttal directly addresses that with new tables. They also criticized metric aggregation and presentation issues; those are only partially addressed. However, I still believe the current empirical evidence is not sufficient.

---

### Decision · Program_Chairs · 2026-01-26

Reject